# Electrochemical Biosensing for Antibiotic-Resistant Bacteria: Advances, Challenges, and Future Directions

**DOI:** 10.3390/mi16090986

**Published:** 2025-08-28

**Authors:** Muhib Ullah Khan, Md. Munibur Rahman, Nusrat Zahan, Mostafa Kamal Masud, Subir Sarker, Md. Hakimul Haque

**Affiliations:** 1Department of Veterinary and Animal Sciences, University of Rajshahi, Rajshahi 6205, Bangladesh; s1810568137@ru.ac.bd (M.U.K.); s1812568108@ru.ac.bd (N.Z.); 2Delta Hospital Limited, Dhaka 1216, Bangladesh; munibmanik@yahoo.com; 3Australian Institute for Bioengineering and Nanotechnology (AIBN), The University of Queensland, Brisbane, QLD 4072, Australia; m.masud@uq.edu.au; 4Biomedical Sciences & Molecular Biology, College of Medicine and Dentistry, James Cook University, Townsville, QLD 4811, Australia; subir.sarker@jcu.edu.au; 5Australian Institute of Tropical Health and Medicine, James Cook University, Townsville, QLD 4811, Australia

**Keywords:** biosensor technology, electrochemistry, antimicrobial susceptibility testing, microfluidic devices, lab-on-a-chip, redox-active metabolites, microbial enzymes, nanotechnology, public health diagnostics

## Abstract

The rapid rise of antibiotic-resistant bacteria (ABR) presents an urgent global health challenge, necessitating the development of efficient and scalable diagnostic technologies. Electrochemical biosensors have emerged as a promising solution, offering high sensitivity, specificity, and adaptability for point-of-care applications. These innovative platforms utilize bio-recognition elements, advanced electrode materials, microbial enzymes, and redox-active metabolites to identify antibiotic resistance profiles at a molecular level. Recent progress in microfluidics and lab-on-a-chip systems has enabled real-time, high-throughput antimicrobial susceptibility testing, significantly improving diagnostic precision and speed. This review aims to critically evaluate recent advances in electrochemical biosensing strategies for detecting ABR, identify key challenges, and propose future directions to enhance clinical applicability. Key developments include bio-receptor-based detection strategies, novel electrode surfaces, and multiplexed platforms integrated with microfluidic systems. Additionally, this review examines essential biomarkers for detecting antibiotic resistance and explores key challenges, including variability in biomarker expression and sensor reproducibility. It also highlights practical barriers to clinical implementation, such as cost constraints and scalability concerns. By presenting innovative approaches, such as cost-effective material alternatives, advanced analytical techniques, and portable biosensing systems, this review outlines a strategic pathway for enhancing the accessibility and effectiveness of electrochemical biosensors in antibiotic resistance management.

## 1. Introduction

Antimicrobial resistance (AMR) has emerged as a critical global challenge affecting public health and agricultural sectors, particularly livestock and poultry management. A key driver of this crisis is the widespread, often indiscriminate, use of antimicrobial agents across various industries [1]. Many of these antimicrobials utilized for growth promotion, disease prevention, and therapeutic applications in animal husbandry, closely resemble or are structurally related to those used in human medicine. Such extensive use accelerates AMR development, shaping its pathobiology through selective pressure and genetic transfer mechanisms. The molecular basis of AMR involves multiple resistance strategies, including modifications to antibiotic target sites, enzymatic degradation, reduced drug permeability, and active efflux systems. These resistance traits can be disseminated through mobile genetic elements such as plasmids, bacteriophages, naked DNA, and transposons, facilitating gene transfer across diverse bacterial species and environmental reservoirs. Moreover, sequential chromosomal mutations driven by antimicrobial selection pressure can escalate bacterial resistance levels [2]. This dynamic exchange of resistance genes among humans, animals, and environmental ecosystems intensifies the spread of AMR, leading to profound ecological disruptions [3,4]. The repercussions of AMR extend far beyond microbial ecosystems, posing significant socioeconomic and healthcare burdens. Rising treatment costs, prolonged hospitalizations, increased morbidity and mortality rates, and clinical complexities affect both developed and developing regions [5]. In resource-limited settings, these challenges are exacerbated by inadequate sanitation, insufficient infection control measures, weak surveillance systems, and the absence of robust antimicrobial stewardship programs [6]. The increasing prevalence of multidrug-resistant (MDR) pathogens underscores the urgent need for effective AMR monitoring strategies. Comprehensive detection methodologies are crucial for environmental surveillance, food safety, and mitigating risks associated with biocorrosion and biodefense applications.

Figure 1 illustrates the timeline and progression of ABR emergence over time. The scientific recognition of AMR dates back to Paul Ehrlich’s observations in 1907, where he documented resistance in trypanosomes, and later, Alexander Fleming’s discovery of penicillin resistance in *Staphylococcus* species in 1945 [7,8]. Since then, numerous traditional methodologies have been employed to detect resistant bacterial strains. Conventional techniques, such as disc diffusion assays, broth microdilution tests, and antimicrobial gradient-based approaches, including Etest strips, rely on bacterial growth patterns in the presence of antibiotics to determine resistance profiles [9]. The advent of automation in clinical microbiology has substantially improved AMR detection efficiency [10]. Regulatory frameworks, such as those established by the European Committee on Antimicrobial Susceptibility Testing (EUCAST) and the Clinical and Laboratory Standards Institute (CLSI), have standardized susceptibility testing protocols, enabling commercial suppliers to provide readily available reagent kits for routine AMR screening [11]. Methods, such as broth microdilution and antimicrobial gradient techniques, yield minimum inhibitory concentration (MIC) values, offering valuable insights into antibiotic effectiveness against bacterial strains [9]. However, these traditional methods often require prolonged incubation periods ranging from several hours to days, limiting their utility in urgent diagnostic scenarios [10]. Additionally, existing approaches frequently suffer from cost constraints, reproducibility challenges, and inadequate sensitivity, which hinder their broader application in environmental and clinical settings [12].

Recognizing the limitations of traditional techniques, researchers have developed an array of innovative approaches to facilitate rapid and precise AMR analysis. Modern molecular detection methods, including polymerase chain reaction (PCR) [11], microarray-based assays [13], whole genome sequencing [14], and mass spectrometry-based techniques [15], have significantly advanced AMR screening capabilities (Figure 2). PCR-based assays, in particular, have revolutionized bacterial resistance detection by directly targeting resistance genes, surpassing conventional culture-dependent techniques in sensitivity and turnaround time [16]. While molecular approaches, such as PCR and nucleic acid sequencing, deliver highly accurate results within 1–4 h, they also pose notable challenges. Sample preparation is often labor-intensive, requiring sophisticated imaging equipment and rigorous contamination controls. Furthermore, these methods detect genetic resistance markers without necessarily correlating them with phenotypic antibiotic susceptibility, limiting the interpretative scope of the findings. Similarly, mass spectrometry (MS)-based techniques, such as matrix-assisted laser desorption ionization-time of flight (MALDI-TOF) MS, have gained traction in clinical microbiology for rapid bacterial identification and resistance profiling. MALDI-TOF MS distinguishes resistant and susceptible bacterial isolates by analyzing the spectral fingerprints of whole cells or crude extracts. Other MS methodologies, such as electrospray ionization (ESI) [17]) and desorption electrospray ionization (DESI) [18], have been employed for AMR characterization in pathogens, including *Escherichia coli*, *Pseudomonas aeruginosa*, *Klebsiella pneumoniae*, *Staphylococcus aureus*, and *Streptococcus agalactiae*. Despite their advantages in speed and automation, mass spectrometry-based techniques face interpretative hurdles, particularly in distinguishing resistance-associated spectral variations from unrelated strain differences. Furthermore, the phenotypic inconsistencies observed in some resistant strains highlight the need for complementary techniques that integrate molecular and functional resistance assessments.

In recent years, there has been a surge in research interest surrounding electrochemical sensing platforms, particularly as alternatives to traditional pathogen detection methods. These sensors offer significant advantages, including high sensitivity, rapid response times, user-friendly operation, and cost-effectiveness [19,20]. Furthermore, electrochemical biosensors have demonstrated versatility in evaluating antibacterial activity [21,22,23,24]. A key advantage of electrochemical sensors lies in their label-free detection capability, which allows for direct bacterial identification without requiring additional reagents or labelling processes. This feature enhances their efficiency and positions them as viable alternatives to conventional diagnostic techniques. Additionally, their ability to support miniaturization makes them suitable for portable, wearable, and implantable applications. These systems have proven effective in detecting whole bacterial cells, signaling molecules, microbial metabolites, enzymatic byproducts, and related biomarkers. Several reviews have explored the scientific principles, technological advancements, and challenges of developing electrochemical biosensing platforms for point-of-care diagnostics [25,26]. For instance, Monzó et al. (2015) provided an extensive evaluation of electrochemical sensor systems, highlighting the potential of microarray-based and microfluidic approaches alongside synthetic polymer integration and cost-effective nanomaterials [27]. Despite these advantages, barriers, such as sample preparation complexities, prolonged analysis times, and sensitivity constraints, have hindered the broader market adoption of these technologies. Kuss et al. (2018) provided a comprehensive analysis of bacteria detection biosensors developed since 2015, emphasizing progress toward cost-effective, highly sensitive, and portable diagnostic solutions [23]. Building on these foundations, Simoska and Stevenson (2019) investigated modern electrochemical strategies for pathogen detection, comparing them with traditional methodologies [20]. Their findings highlighted the utility of micro- and nanoelectrode arrays for real-time bacterial sensing in polymicrobial infections and the integration of microfluidic systems for pathogen separation. More recently, Kim and Yoo (2022) investigated electrochemical sensors for antibiotic susceptibility testing, aiming to enhance sensitivity, enable real-time monitoring, and integrate nanomaterials for improved pathogen detection [28,29]. More recent studies have explored graphene-based and MXene-modified sensors to increase accuracy in identifying antibiotic-resistant bacteria, advancing the development of efficient point-of-care diagnostics [30,31]. Despite these advancements, targeted research on electrochemical biosensors for antibiotic-resistant bacteria detection remains limited. This gap underscores an urgent need for further exploration in this critical area. Figure 1 provides an overview of various detection methods for antibiotic-resistant bacteria documented in the recent literature.

**Figure 2 micromachines-16-00986-f002:**
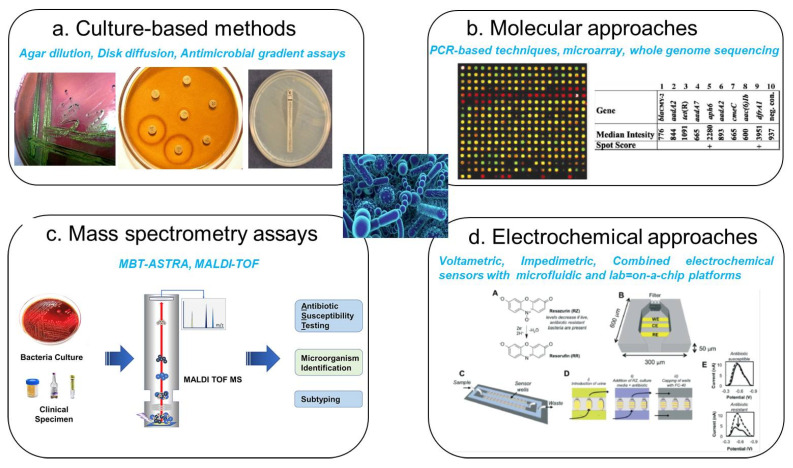
Current techniques for the detection and quantification of antibiotic-resistant bacteria. (**a**) Cell culture methods, including cell plates, disc diffusion assays, and E-test (antimicrobial gradient methods). (**b**) Molecular-based approaches, including nucleic acid sequencing and PCR techniques: image of the *Salmonella* genome AMR gene composite microarray (reprinted with permission [14]. Copyright 2001 American Society for Microbiology). (**c**) Mass spectrometry-based approaches, including ESI, MS imaging, and MALDI-TOF MS: image of bacteria identification based MALDI protein profiling from an unknown bacterial colony (reprinted with permission [16,32]. Copyright 2013 Elsevier). (**d**) Rapid electrochemical phenotypic profiling of antibiotic-resistant bacteria: image is an electrochemical signal of resazurin is attenuated in the presence of viable bacteria, allowing the effects of antibiotics to be assessed electrochemically in 1 h (reprinted with permission [21]. Copyright 2015 the Royal Society of Chemistry).

This review comprehensively assesses recent developments in electrochemical sensing strategies for detecting clinically significant ABR. First, the review examines bio-receptor-based electrochemical sensing approaches that facilitate direct detection of antibiotic-resistant bacteria. Next, it explores novel methodologies utilizing engineered electrode surfaces, redox-active cellular metabolites, and microbial enzymes for enhanced pathogen recognition. Additionally, it highlights multiplexed biosensing platforms that integrate electrochemical detection with microfluidic and lab-on-a-chip technologies to improve diagnostic precision and efficiency. Beyond technological advancements, this review addresses key biological and technical challenges associated with bioelectroanalytical sensors, including biomarker variability, sensitivity, reproducibility, and clinical implementation barriers. The discussion also considers strategies for overcoming current limitations, focusing on innovative solutions such as AI-driven analytics, cost-effective material alternatives, and portable diagnostic platforms. By presenting these perspectives, the review aims to support the advancement of electrochemical biosensors toward practical applications in antibiotic-resistant bacterial detection, fostering their broader adoption in both clinical and environmental settings.

## 2. Clinical and Economic Significance of Detecting Antibiotic-Resistant Bacteria

ABR pose a significant global public health threat, contributing to substantial clinical and economic burdens, particularly for hospitalized patients [6]. The World Health Organization (WHO) has identified six priority antibiotic-resistant pathogens, collectively known as ESKAPE bacteria—*Enterococcus* spp., *Staphylococcus aureus*, *Klebsiella pneumoniae*, *Acinetobacter baumannii*, *Pseudomonas aeruginosa*, and *Enterobacter* spp.—due to their increasing prevalence and role in healthcare-associated infections [33]. These multidrug-resistant organisms complicate treatment strategies, leading to increased morbidity and mortality worldwide. Reports of ESKAPE pathogen outbreaks have been documented across both high- and low-income countries, underscoring the widespread and persistent nature of AMR.

The impact of AMR is alarming, with an estimated 2.8 million infections and approximately 35,000 associated deaths occurring annually in the United States alone [34]. Compromised immunity, often resulting from viral infections, exacerbates the risk of secondary bacterial infections. For example, a systematic review of COVID-19 cases found that 15% of patients developed secondary bacterial infections, accounting for nearly half of the total recorded fatalities [35]. Hospital-acquired infections (HAIs), particularly in intensive care units (ICUs), pose an even greater threat due to the vulnerability of patients with pre-existing conditions, such as diabetes, cancer, and chronic diseases [36]. Given the escalating prevalence of AMR, many national health agencies now recognize it as a critical risk factor warranting inclusion in public health registers. To address this growing concern effectively, advancements in diagnostic technologies are essential for enabling timely, targeted therapy tailored to individual patient needs.

The ability to rapidly and accurately detect antibiotic-resistant bacteria has the potential to revolutionize infection management, allowing for pathogen-specific therapeutic interventions. Point-of-care (POC) diagnostic tools play a pivotal role in improving patient outcomes while mitigating the further development of resistance by ensuring appropriate antibiotic use. In many developing countries, approximately 90% of human antibiotic consumption stems from single prescriptions, underscoring the necessity for advanced diagnostic solutions that guide evidence-based treatment decisions [37]. The UK’s review on AMR, spearheaded by Lord O’Neill, has advocated for measures to curtail antibiotic overuse, emphasizing the integration of rapid diagnostics to support data-driven prescribing practices [38]. Without effective intervention strategies, AMR is projected to cause up to 10 million deaths annually by 2050, with economic losses ranging between $60–100 trillion due to reduced productivity and escalating healthcare costs. While quantifying AMR’s burden in community-acquired infections remains challenging, its devastating effects in nosocomial infections are well documented. In the United States, HAIs account for nearly 99,999 deaths annually, with infection rates averaging 7.1 per 100 patients in Europe and reaching 15.5 per 100 patients in lower-income regions [39]. These infections impose severe financial and societal burdens, leading to preventable illness and mortality. In Europe alone, resistant pathogens contribute to an estimated 25,000 additional deaths per year, with associated healthcare and productivity losses totaling EUR 1.5 billion [40]. Similarly, in the United States, antibiotic-resistant infections result in approximately 2 million cases annually, leading to 23,000 deaths and an estimated USD 30 billion in increased medical expenses and prolonged hospital stays [41].

Given the urgency of combatting AMR, the development of innovative diagnostic technologies capable of rapidly identifying bacterial pathogens and their antibiotic susceptibility profiles at the point of care would be invaluable. An ideal diagnostic platform should be simple to operate, require minimal technical expertise, and deliver immediate results without reliance on complex laboratory infrastructure. By allowing widespread accessibility and efficiency, such advancements could significantly enhance global efforts to monitor, manage, and mitigate the far-reaching consequences of antibiotic resistance.

## 3. Common Biomarkers for Antibiotic-Resistant Bacteria Detection

Antibiotic resistance is a critical global health issue, driven by bacterial adaptations that neutralize the effects of antimicrobial agents. Bacteria employ various resistance mechanisms, including distinct molecular biomarkers such as genes, proteins, and metabolic byproducts that enable them to evade antibiotic effects (Figure 3). Among these biomarkers, beta-lactamase genes (*blaNDM-1*, *blaKPC*, *blaOXA-48*) degrade beta-lactam antibiotics, rendering them ineffective against *Escherichia coli* and *Klebsiella pneumoniae* infections [42]. Likewise, aminoglycoside resistance genes, including *aac (6′)-Ib-cr*, *aph*, and *armA*, neutralise aminoglycosides, interfering with protein synthesis inhibition [43]. The *mecA* gene in methicillin-resistant *Staphylococcus aureus* (MRSA) encodes penicillin-binding protein 2a (PBP2a), reducing the efficacy of beta-lactam antibiotics and facilitating persistent infections [44]. Vancomycin-resistant enterococci (VRE) employ *vanA* and *vanB* genes to modify bacterial cell wall precursors, preventing vancomycin binding and complicating treatment outcomes [45]. Detection methods, such as polymerase chain reaction (PCR), whole-genome sequencing, and biochemical assays, are instrumental in identifying these resistance markers, allowing for timely intervention in clinical and research settings.

Beyond genetic resistance markers, efflux pumps, and metabolic byproducts further complicate antibiotic treatment strategies. Efflux pumps, including MexAB-OprM in *Pseudomonas aeruginosa* and AcrAB-TolC in *Escherichia coli*, actively expel antibiotics from bacterial cells, reducing drug efficacy and driving multidrug resistance [46]. Additionally, quinolone resistance mutations in *gyrA*, *gyrB*, *parC,* and *parE* alter drug targets, preventing quinolones from inhibiting bacterial DNA replication [47]. Metabolic byproducts, such as volatile organic compounds (VOCs), offer alternative diagnostic biomarkers, providing non-invasive detection options through gas chromatography-mass spectrometry (GC-MS) [48]. However, challenges persist due to the broad-spectrum activity of efflux pumps, the variability of VOC profiles, and the rapid emergence of resistance variants, necessitating continuous advancements in molecular detection technologies. Innovative biosensing platforms and refined diagnostic approaches must be developed to combat antibiotic resistance effectively, ensuring early detection and improving antimicrobial stewardship worldwide.

## 4. Electrochemical Detection Techniques of Antibiotic-Resistant Bacteria

In the electrochemical detection of antibiotic-resistant bacteria, a specific biorecognition element (e.g., nucleic acids, peptides, antibodies, aptamers, bacteriophages, etc.) interacts with the target sequence to selectively recognize it, after which a sensitive physical transducer is used to acquire a quantifiable electrochemical signal to measure the level of antibiotic-resistant bacteria [25]. To read the detection, voltametric techniques like cyclic voltammetry (CV), linear sweep voltammetry (LSV), differential pulse voltammetry (DPV), square wave voltammetry (SWV), and stripping voltammetry are used, as well as amperometry and impedimetric methods. Among these readouts, impedimetric and voltametric techniques, especially DPV and SWV, are generally preferred, while amperometry and other methods (e.g., potentiometry) are used less frequently [25]. Because of the versatile ability of electrochemical sensors to detect antibiotic-resistant bacteria with high sensitivity, a wide range of electrochemical assays have been developed in recent years. They are classified into five categories based on their assay construction strategies: *(i) bio-receptors, (ii) electrode materials and modified electrode surfaces, (iii) redox-active cellular metabolites-based assays, (iv) microbial enzymes-based assays, and (v) combined electrochemical sensing techniques (Microfluidic and Lab-on-a-chip based)*. These assays are discussed in detail in the rest of the paper.

### 4.1. Electrochemical Detection of Antibiotic-Resistant Bacteria Based on Bio-Receptors

#### 4.1.1. DNA-Based Sensors

DNA-based electrochemical biosensors function as integrated receptor–transducer systems, where DNA serves as the molecular transducer [49,50]. These sensors translate biochemical interactions, specifically target DNA binding, into measurable electrical signals, enabling rapid and precise detection of genetic markers associated with antibiotic resistance (Figure 4a). The efficiency of DNA-based biosensors depends on the successful immobilization of probe DNA onto the sensor surface, ensuring optimal orientation for hybridization with its complementary target sequence [51]. Effective immobilization techniques are crucial for precise target recognition, as they enhance sensor sensitivity and reliability. Several approaches have been employed to attach probe DNA to sensor electrodes, including adsorption methods, covalent bonding, and avidin–biotin interactions [52]. Each technique plays a role in improving the stability and reactivity of DNA probes within electrochemical sensing systems. Genotypic electrochemical sensors utilize DNA probes derived from genomic DNA, immobilized on electrode surfaces to facilitate the detection of antibiotic-resistance genes or their variants. When the probe DNA hybridizes with its complementary target sequence, it forms a duplex, triggering detectable changes in electrochemical properties such as impedance, current, or capacitance. These variations can be measured using advanced electrochemical techniques, allowing precise quantification of target molecules. One of the key advantages of electrochemical DNA sensors is their high sensitivity, reproducibility, and long shelf life, making them suitable for real-time pathogen detection. Their sensitivity can be further enhanced through signal amplification strategies, including electrode surface optimization and the integration of catalytic components such as enzymes or dye labels. These improvements enable more accurate and efficient bacterial resistance screening, offering a promising alternative to traditional antibiotic susceptibility tests. By leveraging electrochemical DNA biosensors, researchers can advance diagnostic capabilities for antibiotic-resistant bacteria, facilitating more rapid intervention strategies in clinical and environmental settings [29].

#### 4.1.2. mRNA-Based Sensors

Messenger RNA (mRNA)-based detection technologies have emerged as valuable tools for species-specific bacterial identification (Figure 4b). By enabling electrochemical quantification of intracellular mRNA, these methods provide critical insights into bacterial physiology and population dynamics [53]. In 2011, Soleymani and colleagues developed an electrochemical biosensor for bacterial identification using mRNA sequences. Their approach targeted the mRNA encoding the β subunit of RNA polymerase (*rpoB*), a highly expressed and species-specific transcript in bacteria. The sensor utilized thiol-terminated single-stranded DNA probes complementary to *rpoB* mRNA, immobilized on microscopic scaffolds to ensure precise hybridization and detection. The system employed an electrocatalytic assay to measure binding-induced changes, facilitating rapid and accurate bacterial species identification [54]. Another innovative mRNA detection strategy focused on *Escherichia coli*, incorporating nanostructured microelectrodes (NMEs) on silicon-based sensors. Researchers improved target-binding efficiency by enhancing the sensor’s surface area with nanostructured layers of gold or palladium. Peptide nucleic acid (PNA) probes, designed to hybridize with specific *E. coli* mRNA sequences, were immobilized onto the sensor surface using mercaptohexanol (MCH) as a co-ligand to optimize probe density while minimizing non-specific interactions. Bacterial cells were lysed using a rapid cell lysis kit to extract intracellular mRNA for detection. The target mRNA lysate was incubated with the sensor, allowing hybridization with immobilized PNA probes. Detection was performed using differential pulse voltammetry (DPV), coupled with an electrocatalytic reporter system to amplify signal output. Specifically, Ruthenium (III) ions were reduced to Ruthenium (II) during target binding and subsequently regenerated by Ferric (Fe III) ions in a cyclic redox process. This enhanced electrochemical signaling enabled precise *E. coli* mRNA detection [55]. mRNA-based electrochemical biosensors represent a promising alternative for bacterial identification and AMR detection.

**Figure 4 micromachines-16-00986-f004:**
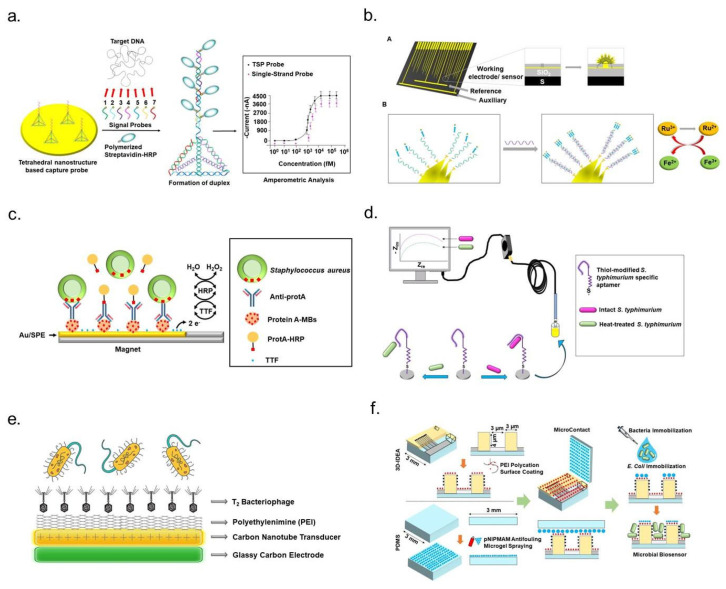
Schematic representation of electrochemical strategies for detecting antibiotic-resistant bacteria. (**a**) DNA-based biosensing: A multisignal probe (MSP) system incorporating seven signal probes and a tetrahedral nanostructure-based capture probe enables electrochemical DNA sensing of *Methicillin-resistant Staphylococcus aureus* (MRSA) through amperometric analysis (reprinted with permission [56]. Copyright 2013 Elsevier). (**b**) mRNA-based detection. (**A**) Illustration of nanostructured microelectrodes fabricated on a silicon chip with integrated gold (Au) working, silver/silver chloride (Ag/AgCl) reference, and platinum (Pt) auxiliary electrodes. Five-micrometer apertures at each working electrode tip serve as electroplating sites, creating substructures approximately 100 μm in diameter. (**B**) Application of ruthenium (III) and iron (III) electrocatalytic reporter pairs to enhance nucleic acid detection (reprinted with permission [57]. Copyright 2012 American Chemical Society). (**c**) Antibody-based sensing: A magnetoimmunosensor designed for *Staphylococcus aureus* employs functionalized magnetic beads (MBs) to capture Staphylococcal protein A (ProtA), concentrating analytes onto tetrathiafulvalene (TTF)-modified gold screen-printed electrodes (Au/SPEs). The addition of hydrogen peroxide (H_2_O_2_) serves as an electrochemical signal transducer (reprinted with permission [58]. Copyright 2012 Springer Nature). (**d**) Aptamer-mediated biosensing: Aptamer-functionalized electrochemical platforms enable selective detection of live *Salmonella typhimurium*, demonstrating a high degree of specificity (reprinted with permission [59]. Copyright 2012 American Chemical Society). (**e**) Bacteriophage-assisted biosensing: A charge-directed approach enables bacteriophage immobilization onto electrode surfaces, enhancing specificity for bacterial detection and analysis (reprinted with permission [60]. Copyright 2022 Royal Society of Chemistry). (**f**) Cell-imprinted electrochemical approach: Bacterial cell-immobilized three-dimensional interdigitated electrode array (3D-IDEA) facilitates impedance-based antibiotic susceptibility testing (AST) for *Escherichia coli* (reprinted with permission [61]. Copyright 2019 Royal Society of Chemistry).

#### 4.1.3. Antibody- and Antibody Fragment-Based Sensors

Antibody-based biosensors, also known as immunosensors, leverage antibodies’ high specificity and strong binding affinity for antigen detection, making them invaluable in biosensing technologies for over two decades. These sensors are categorized into label-free and labelled detection systems, where label-free immunosensors detect biochemical interactions directly (Figure 4c). In contrast, labelled assays use secondary components like enzymes or nanomaterials to generate measurable signals [62,63]. Electrochemical immunosensors employ antibodies or antigens immobilized on electrode surfaces to monitor electrical variations, such as current or voltage changes, resulting from antigen-antibody binding [64,65,66,67]. Structurally, antibodies, or immunoglobulins (Igs) possess a “Y”-shaped configuration comprising two heavy and two light polypeptide chains linked by disulfide bonds, with IgG being the most commonly utilized immunoglobulin due to its stability and strong antigen-binding properties [68]. Immunosensors functioning through antigen-antibody interactions enhance detection precision through electrochemical measurements of impedance, conductivity, or potential shifts, facilitating real-time monitoring of clinically significant analytes. Integrating enzyme-linked or nanoparticle-conjugated antibodies further amplifies detection sensitivity, making these biosensors a robust alternative to conventional diagnostic approaches, particularly for applications requiring rapid and specific pathogen identification [29].

#### 4.1.4. Aptamer-Based Sensors

Aptamers are single-stranded oligonucleotides composed of DNA or RNA that exhibit high specificity and strong binding affinity for various molecular targets, including nucleic acids, proteins, metal ions, and small molecules [69,70]. Their superior sensitivity, selectivity, and stability have made them increasingly preferred over traditional antibodies in biosensing applications. Aptamers, owing to their nucleic acid composition, maintain exceptional structural integrity across diverse environmental conditions, ensuring reliable performance in fluctuating temperatures and storage settings. These biomolecules possess the unique ability to form complex two- and three-dimensional conformations, such as loops, pseudoknots, stems, bulges, hairpins, triplexes, and quadruplexes, facilitating interactions with a broad spectrum of targets, including peptides and small molecules. Additionally, aptamers can be chemically modified to optimize detection efficiency and tailor functionality for specific biosensing applications (Figure 4d). Their remarkable resilience under extreme conditions stems from their rigid nucleic acid backbone and limited conformational flexibility, unlike proteins, which exhibit greater structural variability due to their intricate backbone and side-chain interactions. Aptamers also demonstrate outstanding binding affinity, with dissociation constants typically ranging from the picomolar to the millimolar range. Their small size and structural stability, particularly in the case of DNA aptamers, high specificity and ease of modification, collectively position them as ideal candidates for biosensor development [29]. A compelling demonstration of aptamer-based sensing was presented by Jo et al., who developed a capacitance sensor array functionalized with aptamers to monitor bacterial growth and assess antibiotic susceptibility in *Escherichia coli* and *Staphylococcus aureus* [71]. In this system, DNA aptamers specific to each bacterial strain were immobilized onto gold electrode surfaces, enabling real-time monitoring of bacterial activity. The sensing mechanism was designed such that bacteria function as capacitors arranged in parallel between the electrodes. When antibiotics are applied at inhibitory concentrations, a measurable decrease in capacitance indicates bacterial susceptibility to treatment. Conversely, capacitance increases when bacteria exhibit resistance, mimicking the response observed in untreated bacterial cultures. This innovative approach provides valuable insights into bacterial behavior under antibiotic exposure, facilitating rapid and precise AMR assessment [72].

#### 4.1.5. Bacteriophage-Based Sensors

Bacteriophages, viruses that selectively infect bacterial cells, contain either single- or double-stranded DNA or RNA enclosed within a protective protein capsid. This structural design safeguards their genetic material from environmental damage, allowing them to persist under diverse conditions. Due to their unique biological properties—including high host specificity, the ability to lyse bacterial cells, and replication upon infection—bacteriophages have emerged as powerful tools for biosensing applications aimed at bacterial pathogen detection (Figure 4e) [29]. One notable advancement in bacteriophage-based biosensors was introduced by Harada et al., who developed a detection system targeting antimicrobial-resistant (AMR) bacteria, with a specific focus on *Pseudomonas aeruginosa*, a pathogen notorious for its multidrug resistance [73]. Their approach utilized bacteriophages integrated into a biopolymeric matrix designed for both structural stability and functional efficiency. The researchers engineered two distinct detection systems—one chromogenic and the other bioluminescent—each demonstrating remarkable sensitivity and specificity in identifying *P. aeruginosa*. The chromogenic system incorporated a hydrogel infused with chromogenic substrates that underwent a visible color change upon bacterial lysis. This reaction resulted from interactions between intracellular components released during phage-induced bacterial destruction and the embedded chromogenic agents. In contrast, the bioluminescent system relied on a hydrogel capable of producing luminescence upon bacterial lysis. This luminescent response occurred via a biochemical reaction involving luciferase, luciferin, and ATP, all released from disrupted bacterial cells. Together, these biosensing platforms offer promising solutions for real-time detection of viable AMR *P. aeruginosa* [73]. Beyond their utility in biosensing, bacteriophages hold immense ecological significance as regulators of microbial populations. With an estimated presence exceeding 10^31^–10^33^ globally, they represent one of the most abundant and diverse biological entities in the biosphere. Their resilience to environmental stressors, combined with their ability to rapidly multiply within host bacteria, positions them as critical agents in maintaining microbial equilibrium [74]. These characteristics reinforce their potential for innovative applications in pathogen monitoring and microbial balance assessment [75,76].

#### 4.1.6. Cell- and Molecularly Imprinted Polymer-Based Sensors

Cell-based biosensors provide an innovative approach for AST, enabling the assessment of bacterial responses to antibiotic exposure. A significant advancement in this field involves an impedimetric transducer incorporating a three-dimensional interdigitated electrode array, specifically designed for AST applications (Figure 4f). This system facilitates the spatial immobilization of bacterial cells onto a structured multielectrode platform, which consists of interspersed electrode digits and SiO_2_ insulating barriers. Such a configuration enhances high-throughput detection and analysis capabilities. One of the challenges associated with cell-based biosensors is ensuring consistent bacterial attachment to the electrodes, which is crucial for achieving reproducible results. To mitigate non-specific adhesion, poly (*N*-isopropyl methacrylamide) (pNIPMAM) microgels were employed as antifouling agents, preventing bacterial attachment to the insulating barriers [77]. In this system, *Escherichia coli* cells were electrostatically immobilized onto polyethyleneimine (PEI)-coated polydimethylsiloxane (PDMS) substrates situated on electrode digits. The immobilized bacteria were incubated for 20 min to promote stable attachment. Following antibiotic exposure, a reduction in impedance was observed among antibiotic-susceptible bacteria, likely due to compromised cell membrane integrity. This disruption led to the release of intracellular charged components into the surrounding medium as a consequence of bacterial lysis [78]. Another approach for bacterial immobilization involved integrating *E. coli* cells with graphene to enhance electrochemical sensing capabilities. This method employed a drop-coating technique to disperse bacteria within a graphene-cell mixture onto a modified glassy carbon electrode. The intrinsic oxidoreductase activity of the bacterial cells facilitated electron transfer, enabling real-time evaluation of bacterial viability and antibiotic susceptibility. Differential pulse voltammetry (DPV) was utilized to measure reduction currents after exposure to three antibiotics—ofloxacin, penicillin, and cefepime. The decline in reduction current observed in antibiotic-susceptible cells indicated a loss of metabolic activity. Conversely, antibiotic-resistant strains maintained electrochemical reduction current intensity, reflecting continued bacterial viability despite antibiotic treatment [28]. These biosensing strategies highlight the potential of cell-based detection systems in AST applications, offering high specificity and sensitivity for monitoring bacterial responses to antimicrobial agents.

#### 4.1.7. Carbohydrate-Based Sensors

D-mannose (D-MAN), a hexose sugar classified under the aldohexose group, is widely present in human epithelial cells and plays a significant role in biosensing applications. A straightforward method has been developed to immobilize D-MAN on the surface of glassy carbon electrodes (GCEs), enabling its use as a receptor for bacterial detection. In this system, D-MAN was immobilized on gold nanoparticle (AuNP)-modified GCEs to facilitate Escherichia coli recognition. A notable carbohydrate-based electrochemical biosensor was designed by Hargol Zadeh et al., integrating p-carboxyphenylamino mannose (PCAM) covalently attached to gold nanoparticles electrodeposited onto a GCE surface. This biorecognition layer enabled rapid and highly sensitive bacterial detection, demonstrating a strong linear correlation (R^2^ = 0.998) between bacterial concentration (log scale) and sensor response across a range of 1.3 × 10^1^ to 1.3 × 10^6^ colony-forming units per milliliter (CFU/mL). The biosensor achieved an impressive detection limit of 2 CFU/mL within 60 min, highlighting its capability for rapid bacterial monitoring [79]. Another innovative carbohydrate-based biosensor was developed by Saucedo et al., employing a conductive polymer transducer for bacterial detection. This system utilized a 4-(3-pyrrolyl) butyric acid film electrochemically deposited onto a gold electrode, which was subsequently functionalized with lectins acting as bioreceptors. The biosensor operated by detecting viable bacteria through the production of acidic metabolites in response to glucose exposure. The metabolic activity of bacterial cells led to a potential shift due to the accumulation of acidic metabolites. Additionally, this biosensor demonstrated its potential in assessing antibiotic efficacy by monitoring signal variations upon bacterial exposure to antibiotics. A decrease in metabolite production indicated bacterial death or diminished activity, providing insight into antibiotic effectiveness. The biosensor was tested with live *E. coli* cells exposed to tetracycline, erythromycin, and kanamycin at a concentration of 4 mg/mL. Results confirmed that tetracycline and erythromycin significantly inhibited *E. coli*, leading to a notable reduction in signal intensity, whereas kanamycin exhibited limited efficacy, producing a higher residual signal [26]. These findings underscore the potential of carbohydrate-functionalized biosensors for bacterial detection and antimicrobial susceptibility testing, offering high sensitivity and specificity for real-time monitoring.

### 4.2. Electrochemical Detection of Antibiotic-Resistant Bacteria Based on Electrode Materials and Modified Electrode Surfaces

#### 4.2.1. Carbon-Based Sensors (Carbon Nanotubes and Graphene)

Graphene, a unique two-dimensional nanomaterial, exhibits exceptional physicochemical properties that make it highly advantageous for electrochemical sensor electrodes, particularly in detecting antibiotic-resistant bacteria. Its versatility has been demonstrated across various biosensing applications aimed at identifying bacterial pathogens. One significant development in graphene-based sensors was reported by Wang et al., who utilized a reduced graphene oxide (rGO)-modified glassy carbon electrode as a label-free detection platform for methicillin-resistant Staphylococcus aureus (MRSA). This system employed electrochemical impedance spectroscopy (EIS) to achieve highly sensitive bacterial detection (Figure 5a) [80]. Further advancements in graphene-based biosensing include the work of Kumar et al., who developed a highly sensitive electrical detection system using graphene field-effect transistors (G-FETs). Their approach enabled rapid, selective, single-cell detection of antibiotic-resistant bacteria through pyrene-conjugated peptides immobilized on G-FETs. These peptides were engineered for precise recognition of pathogenic *S. aureus* strains [81]. Additionally, a similar graphene-based detection strategy has been explored for distinguishing antibiotic-resistant strains of *Acinetobacter baumannii* from their susceptible counterparts. To optimize bacterial adhesion to the graphene transducer, an electric-field-assisted binding technique was incorporated. This method involved applying electrical pulses to direct bacterial cells toward the graphene surface, significantly improving detection efficiency [72].

#### 4.2.2. Paper-Based Sensors

Paper-based sensors provide a cost-effective, portable solution for detecting antibiotic-resistant bacteria, such as *Escherichia coli*, and can be adapted to integrate pNIPMAM microgels as antifouling agents, PEI-coated PDMS for electrostatic bacterial immobilization, and impedance-based detection of antibiotic susceptibility (Figure 6) [85]. These systems, exemplified by colorimetric biosensors detecting β-lactamase activity using nitrocefin [86] or electrochemical sensors incorporating polyaniline-coated paper with silver electrodes [87], enable rapid pathogen detection via colorimetric changes or impedance shifts. To improve specificity, pNIPMAM microgels can be applied to paper substrates, preventing non-specific bacterial adhesion [88], while PEI functionalization can facilitate electrostatic immobilization of *E. coli* [77]. The principle of impedance-based bacterial detection aligns with existing methods assessing membrane disruption and intracellular component release following antibiotic exposure [78], providing a foundation for the development of point-of-care diagnostics using screen-printed electrodes. Furthermore, paper-based sensors can be adapted for genetic detection, including antibiotic-resistant gene mutations, by integrating isothermal amplification techniques with electrochemical readouts. These systems employ a screen-printed gold electrode functionalized with oligonucleotides acting as capture probes and primers, alongside reagents, such as deoxynucleotide triphosphates (dNTPs), recombinase, polymerase, and horseradish peroxidase (HRP)-labeled dTTP, to facilitate amplification. The process involves recombinase binding primers to their complementary sequences within double-stranded DNA, unwinding the helix without requiring thermal cycling, followed by polymerase-mediated strand synthesis incorporating HRP-labeled dTTP. Upon introducing hydrogen peroxide (H_2_O_2_) and 3,3′,5,5′-tetramethylbenzidine (TMB), HRP catalyzes the oxidation of TMB, generating an amperometric signal proportional to the quantity of target DNA. This approach achieves exceptional sensitivity, detecting oxacillin-resistant DNA at concentrations as low as 0.69 femtomolar (fM), corresponding to 319 colony-forming units per milliliter (CFU/mL) or 10.6 attomolar (aM) of target DNA [28], underscoring the potential of paper-based biosensors for real-time, point-of-care diagnostics in AMR detection. Recently, Aurchey et al. developed a low-cost, rapid PASMAT device for antibiotic susceptibility testing, utilizing a colorimetric, length-based zone of inhibition to evaluate bacterial resistance. The device demonstrated high accuracy (up to 100% agreement with CLSI standards), faster results within 8 h, and significantly reduced costs compared to traditional methods [89].

#### 4.2.3. Nanoparticle-Based Sensors

Nanoparticle-based biosensors have transformed the detection of antibiotic-resistant bacteria, including *Escherichia coli*, by utilizing functional nanomaterials to enhance sensitivity, specificity, and speed in AST (Figure 5c). These sensors offer a promising adaptation for impedance-based detection systems incorporating pNIPMAM microgels as antifouling agents and PEI-coated PDMS for electrostatic bacterial immobilization. A wide range of nanomaterials, such as metal oxides, carbon-based nanomaterials, and magnetic nanoparticles, are employed to functionalize electrodes, significantly improving electrochemical response through enhanced electron transfer kinetics, catalytic activity, and biocompatibility [29]. For instance, nanoscale cavities formed on electrode surfaces establish microenvironments that amplify detection signals, while nanoimpact electrochemistry facilitates single-entity detection via faradaic charge transfer during nanoparticle-electrode collisions, markedly increasing sensitivity [28]. Within the described system, integrating pNIPMAM microgels with nanoparticle-functionalized electrodes could mitigate non-specific bacterial adhesion, and PEI-modified surfaces could optimize electrostatic bacterial immobilization on nanomaterial-coated substrates. These advancements align with impedance-based methods for monitoring membrane disruption and intracellular component release upon antibiotic exposure [78]. The synergy of nanoparticle-enhanced biosensing strategies offers a highly precise, rapid, and portable diagnostic platform, facilitating improved antibiotic stewardship by reducing unnecessary prescriptions and advancing AMR monitoring in clinical and environmental settings.

### 4.3. Electrochemical Detection of Antibiotic-Resistant Bacteria Based on Redox-Active Cellular Metabolites

Electrochemical biosensors offer a fast and sensitive method for detecting antibiotic-resistant bacteria, such as *Escherichia coli*, by measuring redox-active cellular metabolites like resazurin and pyocyanin (PYO), which reflect bacterial activity and resistance. Redox indicators, such as resazurin, are enzymatically reduced by metabolically active bacteria to resorufin, causing detectable changes in electrical current (Table 1) (Figure 5d) [92], while PYO secretion from Pseudomonas aeruginosa biofilms can be monitored using Square-Wave Voltammetry to assess resistance to antibiotics like colistin sulfate in real time [93]. These sensors can perform AST in under 30 min without the need for bacterial cultivation. To improve accuracy, antifouling pNIPMAM microgels are used to prevent non-specific bacterial adhesion, and PEI-coated PDMS surfaces enhance bacterial immobilization through electrostatic interactions [78]. Additionally, nanoparticle-enhanced electrodes, including those functionalized with carbon nanotubes, significantly improve sensitivity, enabling precise detection of redox-active metabolites and supporting impedance method that monitor bacterial membrane disruption and intracellular leakage after antibiotic exposure [94].

### 4.4. Electrochemical Detection of Antibiotic-Resistant Bacteria Based on Microbial Enzymes

Electrochemical biosensors leveraging microbial enzyme activity have significantly advanced the detection of ABR, providing a rapid and sensitive approach for identifying resistant strains. These biosensors utilize enzymes—biological catalysts—as biomarkers to differentiate bacterial species based on their metabolic interactions with specific substrates. The detection mechanism relies on either enzymatic substrate conversion or inhibition of enzymatic activity, enabling precise identification of bacterial pathogens. In recent years, enzymatic biosensors have gained prominence in food safety applications, particularly for detecting pathogenic bacteria linked to foodborne diseases [118]. A pivotal advancement in this field is the electrochemical detection of carbapenemase enzymes, a subclass of β-lactamases responsible for hydrolyzing a broad spectrum of antibiotics, significantly contributing to multidrug resistance. The presence of these enzymes serves as a critical indicator of emerging antibiotic resistance within bacterial populations. Genes encoding carbapenemases are widely disseminated among members of the *Enterobacteriaceae* family, *Pseudomonas*, and *Acinetobacter* species, predominantly through horizontal gene transfer, exacerbating the global challenge of drug-resistant infections [119]. Electrochemical biosensors have been successfully employed to detect carbapenem resistance genes. For instance, Huang et al. (2015) developed a biosensor integrating gold screen-printed electrodes (SPEs) with electrochemical impedance spectroscopy (EIS) to identify the New Delhi metallo-β-lactamase (NDM) gene (Table 1) [120]. Similarly, Pan et al. (2015) introduced an electrochemical DNA biosensor capable of detecting the *Klebsiella pneumoniae* carbapenemase (KPC) gene [121]. Expanding on these developments, Subak et al. (2024) proposed an innovative electrochemical nanobiosensor targeting antibiotic resistance genes, specifically OXA-48-type oxacillinase and Verona integron-encoded metallo-β-lactamase (VIM), which contribute to carbapenem resistance [122]. This groundbreaking diagnostic kit employs symmetric and asymmetric PCR techniques as a modern alternative to conventional agarose gel electrophoresis. To enhance sensor performance, the system integrates multiwalled carbon nanotubes (MWCNTs), strengthening the interaction between the sensing surface and the electrode via cyclic voltammetry (CV). Surface passivation strategies further improve sensor stability, ensuring functional reliability for at least 150 days. This highly specialized platform presents a robust and scalable tool for detecting carbapenemase-mediated antibiotic resistance, with promising applications in clinical diagnostics and epidemiological surveillance [122].

### 4.5. Electrochemical Detection of Antibiotic-Resistant Bacteria Based on Combined Electrochemical Sensing Techniques

#### 4.5.1. Microfluidic Based

Microfluidic biosensors have revolutionized the electrochemical detection of antibiotic-resistant bacteria, such as *E. coli*, by offering rapid, highly sensitive, and portable AST (Figure 5b). These platforms complement impedance-based detection strategies that incorporate (pNIPMAM microgels to reduce non-specific bacterial adhesion and PEI-coated PDMS for electrostatic bacterial immobilization. A notable microfluidic biosensing system developed by Yang et al. (2020) utilizes PDMS-based microchannels with nanoconstrictions to isolate individual bacterial cells, enabling all-electrical detection of multidrug-resistant strains within two hours by monitoring electrical resistance variations during antibiotic exposure [123]. Integrating pNIPMAM microgels into this system could further enhance specificity by minimizing non-specific interactions on electrode surfaces, while PEI functionalization would improve bacterial immobilization, aligning with impedance-based methods that assess membrane integrity and intracellular component leakage following antibiotic treatment [78]. Recent advances in microfluidic electrochemical sensors, particularly those incorporating nanostructured electrodes, have significantly improved detection sensitivity and assay efficiency. These innovations support the development of high-throughput diagnostics capable of real-time bacterial susceptibility analysis, providing essential tools for effective antibiotic stewardship and resistance monitoring [124,125,126]. The integration of these biosensing technologies underscores their potential for scalable, point-of-care applications, facilitating rapid and precise detection of antibiotic-resistant pathogens in both clinical and environmental settings.

#### 4.5.2. Lab-on-a-Chip Based

Lab-on-a-chip (LOC) platforms have revolutionized AST by integrating electrochemical sensing and analytical precision into miniaturized, portable devices. These advanced biosensing systems complement impedance-based detection strategies incorporating pNIPMAM microgels to mitigate non-specific bacterial adhesion and PEI-coated PDMS for electrostatic bacterial immobilization. A notable LOC biosensor described by Kim and Yoo (2022) employs antibody-functionalized microchips with carbon-coated substrates and silver interdigitated electrodes, enabling efficient bacterial capture from complex clinical samples, such as blood and urine [28]. This system facilitates impedance-based AST within 90 min, providing a rapid assessment of bacterial resistance to antibiotics such as ampicillin and ciprofloxacin. Integrating pNIPMAM microgels into LOC systems could further refine specificity by preventing unwanted surface interactions, while PEI-functionalized coatings enhance bacterial immobilization, ensuring accurate impedance-based detection of membrane disruption and intracellular leakage following antibiotic exposure [78]. Furthermore, high-throughput LOC designs featuring nanostructured electrodes offer enhanced sensitivity and scalability, positioning these biosensors as ideal solutions for real-time, point-of-care diagnostics. Recent advances in LOC technology have contributed to optimizing antibiotic stewardship by enabling precise and rapid detection of resistant pathogens, thereby supporting global efforts to combat AMR [127].

## 5. Biological Challenges in Electrochemical Detection of Antibiotic-Resistant Bacteria

### 5.1. Influence of Sample Matrix on Assay Reliability

The inherent complexity and variability of biological samples pose significant challenges for the consistency and accuracy of electrochemical assays. Factors, such as pH fluctuations, nutrient composition, bacterial diversity, and inoculum size, can drastically impact assay performance, particularly in clinical settings. For example, ascorbic acid in urine has been shown to interfere with resazurin-based assays, potentially resulting in false-negative readings [128]. Similarly, variations in pH can influence the stability of pH-sensitive dyes, such as phenol red, which are widely utilized for bacterial growth detection. These inconsistencies underscore the need for robust assay optimization to ensure the reliability of AST across diverse sample matrices [129,130].

### 5.2. Challenges Posed by Polymicrobial Infections

Polymicrobial infections present unique diagnostic hurdles, as conventional electrochemical assays are often designed for single-colony isolates and may lack the capability to differentiate multiple bacterial species within a single sample. For instance, mixed urinary tract infections (UTIs) frequently involve a dominant pathogen coexisting with commensal bacteria, complicating pathogen identification and resistance profiling. Accurate diagnosis in such cases necessitates advanced biosensing approaches capable of distinguishing and quantifying distinct microbial populations within polymicrobial communities [131].

### 5.3. Impact of Inoculum Size on Antibiotic Susceptibility Testing

The bacterial concentration in clinical samples plays a crucial role in AST outcomes, with higher bacterial loads often diminishing antibiotic efficacy—a phenomenon known as the inoculum effect (IE). This effect is particularly pronounced in beta-lactam resistance assays, where increased inoculum sizes can lead to misleading susceptibility categorizations [132]. Addressing this issue requires standardization techniques such as inoculum normalization via filtration or dilution to enhance assay reproducibility and prevent inaccurate resistance classification [133].

## 6. Technical Challenges in Electrochemical Detection of Antibiotic-Resistant Bacteria

### 6.1. Growth Media Compatibility and Nutritional Stability

Maintaining standardized nutritional conditions is essential for bacterial cultivation and reliable AST. Even minor variations in growth media composition can influence assay accuracy. For instance, MRS medium is optimized for lactic acid bacteria, whereas ISA media is better suited for other bacterial species, underscoring the need for careful media selection to ensure reproducible results [134]. To mitigate inconsistencies, strategies such as controlled diffusion, metabolic stability, and regulated antibiotic elimination during AST procedures must be implemented.

### 6.2. Limitations in Molecular Biology Approaches

Molecular biology techniques, including recombinant expression and genetic analysis, have significantly enhanced detection sensitivity but present notable constraints. Identifying resistance genes requires specialized expertise and access to advanced instrumentation, which may not be feasible in all settings. Moreover, genetic markers may not universally apply to all clinically relevant pathogens, necessitating species-specific approaches. Fluorescent dyes, often used to improve detection simplicity, require high-resolution imaging tools, limiting their practical application in resource-limited environments [135].

### 6.3. Challenges of Non-Specific Biomolecule Interactions

Electrochemical biosensors are highly susceptible to non-specific binding of biomolecules, such as proteins, within complex biological samples. This interference can lead to false-positive readings, reducing diagnostic precision. To counteract non-specific interactions, blocking agents like mercaptohexanol, polyethylene glycol, and bovine serum albumin are employed; however, optimizing these methods to ensure reproducibility and minimize background noise remains an ongoing challenge.

### 6.4. Sensitivity Constraints in Miniaturized Platforms

Microfluidic biosensors designed for single-cell analysis enable rapid bacterial detection, yet the miniaturization process introduces limitations in analytical sensitivity. Lower sample volumes may restrict bacterial availability, affecting overall detection efficiency. For example, a single colony-forming unit (CFU) within a 1 µL sample might go undetected if bacterial concentrations are too low, necessitating alternative strategies, such as increasing sample input or integrating bacterial trapping mechanisms, to enhance assay performance [136].

## 7. Potential Strategies and Solutions for Addressing Challenges in Electrochemical Detection of Antibiotic-Resistant Bacteria

### 7.1. Improving Sample Matrix Compatibility

The variability of biological sample matrices poses a significant challenge in electrochemical assays, necessitating standardized sample preparation protocols. Key approaches include pH adjustment, interfering substances removal, and bacterial concentration normalization to enhance assay reliability. The incorporation of specialized filters and neutralizing agents can mitigate the effects of pH fluctuations and enzymatic activity, particularly in complex samples such as blood and urine. Furthermore, the development of enhanced indicators, such as modified resazurin derivatives, can improve assay sensitivity by reducing interference from endogenous substances.

### 7.2. Addressing Polymicrobial Infections

Polymicrobial infections complicate bacterial detection and antibiotic susceptibility testing (AST), as different species may interact and influence each other’s growth and resistance mechanisms [137]. Species-specific biomarkers and DNA probes enable multiplexed detection, allowing the differentiation of bacterial species within a single assay [138]. Fluorescence microscopy and spectral imaging techniques further aid in visualizing and characterizing bacterial communities in polymicrobial samples [139]. Advanced genetic tools, including quantitative PCR (qPCR) and next-generation sequencing (NGS), provide rapid and precise identification of resistant pathogens, facilitating targeted treatment strategies.

### 7.3. Mitigating the Inoculum Effect

The inoculum effect where high bacterial densities reduce antibiotic efficacy poses a challenge for AST. Automated diagnostic systems integrating real-time bacterial growth monitoring with filtration and dilution techniques can normalize inoculum size and enhance test reproducibility. Additionally, dynamic antibiotic dosing strategies in microfluidic platforms allow for adaptive adjustments in response to bacterial proliferation, improving the accuracy of susceptibility results [140]. Pre-enrichment methods, such as selective culturing or immunomagnetic separation, offer further improvements in standardizing bacterial loads without requiring lengthy overnight incubation (Pre-enrichment methods, such as selective culturing or immunomagnetic separation, offer further improvements in standardizing bacterial loads without requiring lengthy overnight incubation [141].

### 7.4. Enhancing Growth Media Performance

Optimizing bacterial growth media is crucial for reliable AST, particularly for slow-growing or fastidious pathogens [142]. The development of synthetic growth media tailored for specific bacterial species can enhance growth consistency and reduce variability across different assays. Microfluidic platforms further contribute by maintaining stable environmental conditions and minimizing evaporation effects. Integrating pharmacokinetic and pharmacodynamic (PK/PD) models into AST can improve clinical relevance by replicating antibiotic behavior in vivo, thereby enhancing predictive accuracy for treatment efficacy [143].

### 7.5. Use of Functional Materials

Material selection is crucial for point-of-care biosensors, affecting cost, sensitivity, stability, and usability. Rigid materials like silicon, glass, and quartz support detection platforms, while flexible options, such as PDMS (poly (dimethylsiloxane)), PVA (polyvinyl alcohol), PEN (polyethylene naphthalate), and paper, enable miniaturization and cost-efficiency. Paper-based systems, especially PDMS hybrids, offer improved stability for bacterial detection [144]. Polymers like EVA provide biocompatibility and hydrophobicity for microfluidics, while carbon-based materials, like nanotubes [145] and graphene [146], enhance signal transmission and efficiency in biosensing.

### 7.6. Leveraging Molecular Biology Tools

Combining molecular diagnostics with conventional AST enhances precision and reduces turnaround times. Portable, cost-effective genomic platforms, such as CRISPR-based diagnostic tools and isothermal amplification techniques, enable rapid identification of resistance genes in point-of-care settings [131]. Moreover, advancements in fluorescent dyes and nanoprobes with high specificity improve signal-to-noise ratios, reducing false positives and enhancing the reliability of molecular detection [147].

### 7.7. Minimising Non-Specific Responses

Biosensor accuracy can be compromised by non-specific interactions, leading to false-positive or ambiguous results. Strategies, such as coating sensor surfaces with anti-fouling polymers and incorporating blocking agents, help reduce non-target adsorption [148]. Signal amplification methods, including enzyme-linked detection and electrochemical impedance spectroscopy, further enhance assay specificity [149]. Additionally, multilayered sensor architectures incorporating selective membranes can filter out unwanted molecules while ensuring high-affinity target detection [150].

### 7.8. Maximizing Sensitivity in Miniaturized Systems

Miniaturized diagnostic platforms face sensitivity limitations due to reduced sample volumes. Pre-concentration techniques, such as centrifugation and filtration, can enhance bacterial detection by isolating and concentrating target cells before analysis [151]. Microfluidic designs incorporating bacterial trapping mechanisms and on-chip enrichment strategies further improve detection efficiency [138]. Single-cell analysis tools, including digital PCR and microarray-based platforms, provide high-resolution tracking of bacterial growth, enabling earlier and more precise infection detection [152].

### 7.9. Overcoming Logistical and Cost Constraints

The widespread implementation of electrochemical diagnostic tools requires cost-effective and scalable solutions. The development of disposable, paper-based biosensors offer an affordable alternative for resource-limited settings while maintaining diagnostic accuracy. Modular diagnostic platforms that integrate with existing healthcare infrastructure can facilitate widespread adoption across diverse clinical environments [153]. Additionally, fostering collaborations between academic institutions, industry partners, and healthcare providers is critical for accelerating the translation of innovative diagnostic technologies into clinical practice.

By addressing these challenges through innovative strategies, electrochemical detection methods can achieve greater accuracy, reliability, and accessibility. These advancements are crucial for mitigating the impact of antibiotic resistance and enhancing global public health efforts.

## 8. Conclusions and Future Perspectives

Electrochemical biosensors offer a transformative solution for tackling the escalating challenge of ABR. By providing rapid, sensitive, and cost-effective alternatives to conventional diagnostic methods, these biosensing technologies have the potential to revolutionize AMR detection. Significant advancements, such as nanomaterial-functionalized electrodes, microfluidic platforms, and lab-on-a-chip systems, have enhanced detection capabilities, improving sensitivity, specificity, and adaptability across diverse applications. However, key challenges continue to limit widespread implementation, including the complexity of biological samples, the need for polymicrobial detection, and ensuring consistent performance in varied clinical and environmental conditions. Future research should focus on developing more robust and multifunctional biosensors capable of simultaneously detecting multiple resistance markers, addressing the heterogeneity of antibiotic resistance mechanisms. The integration of advanced nanotechnologies will further refine sensor performance, while artificial intelligence and machine learning algorithms could enable real-time data interpretation, enhancing diagnostic accuracy and predictive modeling. Additionally, the development of miniaturized and portable biosensors will be crucial for expanding point-of-care applications, particularly in resource-limited settings where access to conventional laboratory infrastructure is constrained. Standardization of fabrication techniques and analytical protocols is essential to ensure reproducibility and regulatory compliance across various sectors. Moreover, the exploration of sustainable materials and scalable manufacturing strategies will play a pivotal role in making these technologies more accessible and environmentally responsible. To guide future efforts, this review proposes a strategic roadmap, including short-term goals, such as developing standardized validation protocols, to improve reproducibility, medium-term goals, like initiating clinical trials to assess biosensor performance in real-world settings, and long-term goals focused on achieving widespread adoption in point-of-care diagnostics through scalable manufacturing and regulatory approval. Electrochemical biosensors can significantly contribute to combating AMR, improving clinical outcomes, and reinforcing global public health initiatives through continuous interdisciplinary collaboration and technological innovation.

## Figures and Tables

**Figure 1 micromachines-16-00986-f001:**
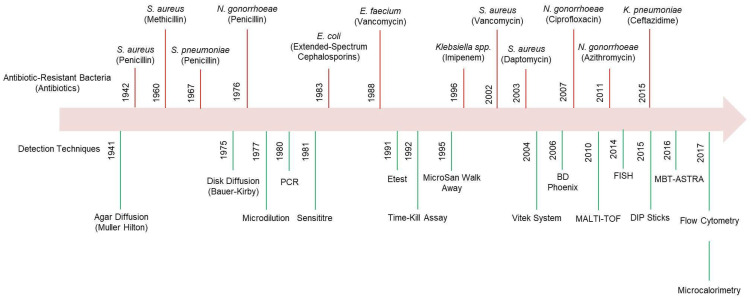
Development of ABR across multiple generations of antibiotics, alongside the evolution of detection tools used to identify AMR. MALDI-TOF—matrix-assisted laser desorption/ionization time-of-flight; FISH—fluorescence in situ hybridization; MBT-ASTRA—MALDI Biotyper antibiotic susceptibility test rapid assay; PCR—polymerase chain reaction.

**Figure 3 micromachines-16-00986-f003:**
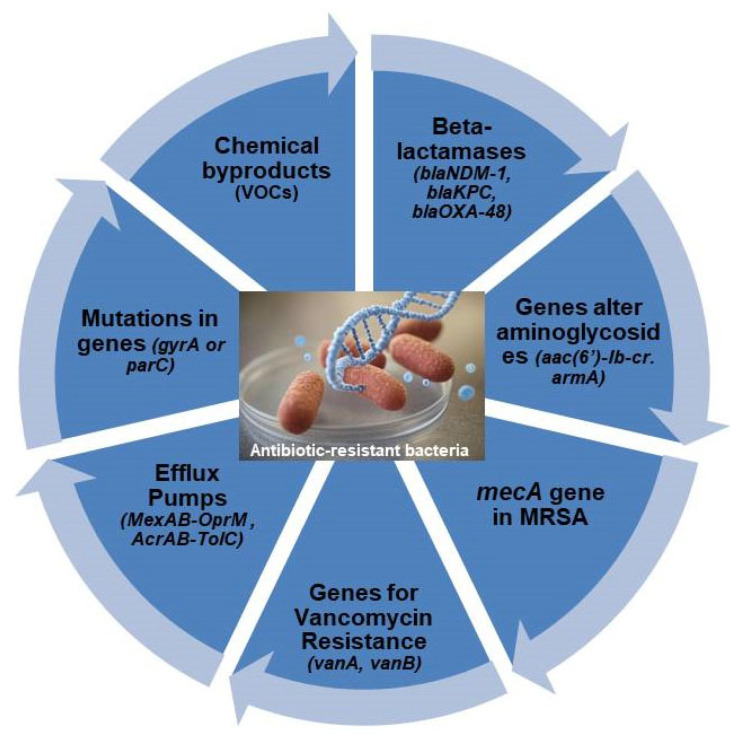
Common biomarkers for detecting Antibiotic-Resistant Bacteria.

**Figure 5 micromachines-16-00986-f005:**
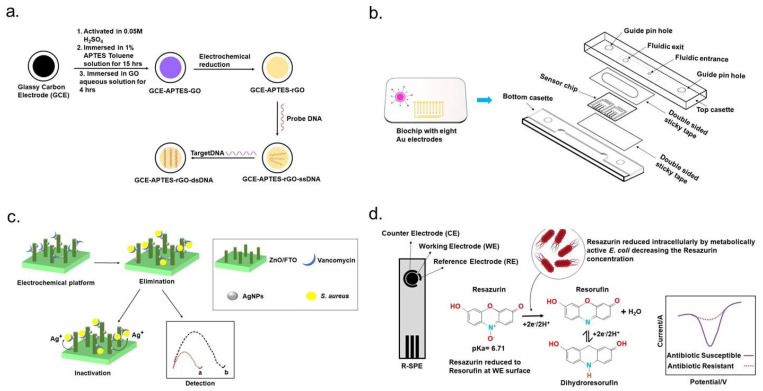
Schematic illustration of electrochemical strategies for detecting antibiotic-resistant bacteria. (**a**) Graphene-based biosensing: Functionalized glassy carbon electrodes enable DNA-based detection, while a graphene field-effect transistor (G-FET) with a pyrene-conjugated peptide probe interacts with bacterial surfaces. A light microscopy inset highlights the G-FET’s active area (10 × 40 μm) positioned between gold contacts (reprinted with permission [80]. Copyright 2013 Elsevier). (**b**) Microfluidic-based biosensing: A fully automated microfluidic electrochemical sensor facilitates real-time amperometric monitoring, streamlining pathogen detection with high precision (reprinted with permission [82]. Copyright 2013 Elsevier). (**c**) Nanomaterial-enhanced biosensing: Silver nanoparticle (AgNP)-decorated ZnO nanorod platforms leverage impedance spectroscopy for simultaneous bacterial identification, eradication, and inactivation, specifically targeting *Staphylococcus aureus* (reprinted with permission [83]. Copyright 2013 Elsevier). (**d**) Redox-active biosensing: Phenotypic antibiotic susceptibility testing (AST) employs an electrochemical method using resazurin. The dye undergoes intracellular reduction by metabolically active bacteria or direct electrochemical transformation at the electrode surface, distinguishing antibiotic-susceptible strains from resistant ones (reprinted with permission [84]. Copyright 2021 Royal Society of Chemistry).

**Figure 6 micromachines-16-00986-f006:**
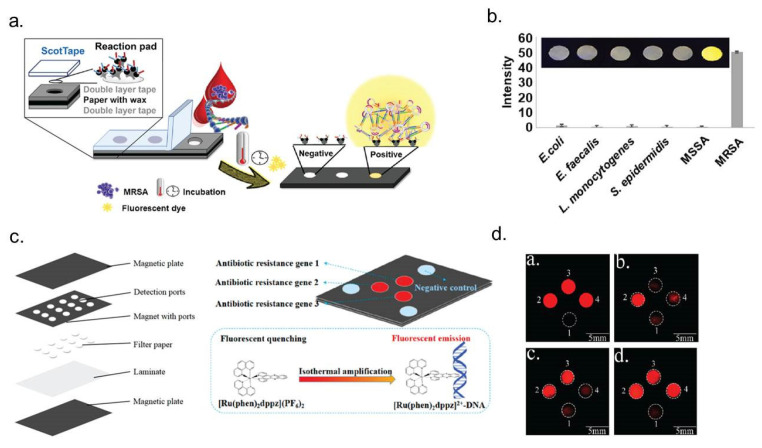
Schematic representation of paper-based LAMP (loop-mediated isothermal amplification) with fluorescence detection for identifying antibiotic-resistant bacteria. (**a**) Paper-based LAMP device reaction pad containing LAMP reagents, fluorescent dye, and the target sample (MRSA). Reprinted with permission from [90]. Copyright 2021 American Chemical Society. (**b**) Fluorescence signal indicating MRSA-positive samples using the paper-based LAMP device. Reprinted with permission from [90]. Copyright 2021American Chemical Society. (**c**) Paper-based chips consisting of five layers: magnetic plate, magnetic ports, filter paper, laminate, and magnetic plates. Isothermal amplification triggers interaction between the metal complex and antibiotic-resistant genes, generating a fluorescence signal. Reprinted with permission from [91]. Copyright 2018 American Chemical Society. (**d**) Multiplex detection of antibiotic-resistant genes: spot 2 (16S rDNA), spot 3 (ermC), spot 4 (mecA), and spot 1 as a negative control. Reprinted with permission from [91]. Copyright 2018 American Chemical Society.

**Table 1 micromachines-16-00986-t001:** Application of electrochemical sensors used in antibiotic resistance evaluation.

**Method**	Working Electrode	Probe	Target Pathogens	Tested Antibiotics	Assay Time	Linear Range	Detection Limit	Advantages and Limitations	**Ref.**
Square wave voltammetry (SWV)	Gold electrode	Solid-phase isothermal primer	*Mycobacterium tuberculosis*	Rifampicin	25 min	6 µM–140 µM	3 pM	Rapid approach and detecting multiple SNPs; additional development and validation may require	[95]
Chronoamperometry	SPGE	HRP-labeled thymine nucleotides	*E. coli*	Oxacillin	60 min	208 × 10^3^ to 2.08 × 10^3^ cfu/mL	319 CFU/mL	Rapid detection; direct coupling of the modified RPA reaction to electrochemical measurement simplifies the detection process	[96]
Cyclic voltammetry (CV)	L-lysine coated CeO/ITO	K_3_Fe [CN]_6_	*E. coli*	Ciprofloxacin, cefixime, amoxicillin	15 min	1.07 × 10^5^ to 1.3 × 10^8^ CFU/mL	4.5×10^4^ CFU/mL	Faster than traditional AST method	[97]
Photoclectrochemistry	S_DNA1_-SbT@SiO_2_NSs complex and Sb_2_S_3_/ZnS/ITO	DNA	*E. coli*	Penicillin	Not reported	1 nM to 10 µM	1 nM	It can be successfully used for measuring *bla_-CTX-M-1_* and *bla_-TEM_* in real *E. coli* plasmids	[98]
Impedance	Gold electrode with surface graphene ink	ssDNA-GE	b-lactam gene	Ampicillin	1 h	6.3–900 ng/mL	6.3 ng/mL	Specific recognition ability for single-base, double-base, and three-base mismatch DNA	[99]
Impedance	Tantalum silicide electrode	PEI/p(NIPMAM/PDMS microgel	*E. coli*	Ampicillin	60–120 min	2–8 mg/L	2 mg/L	High sensitivity, real-time monitoring; requires complex sample preparation	[59]
Impedance	Silver interdigitated carbon working electrode, counter electrode, and reference electrode	Label free	*E. coli*, Methicillin-resistant *Staphylococcus aureus* (MRSA)	Ampicillin, erythromycin, ciprofloxacin, methicillin, daptomycin, gentamicin	<90 min	0.1 µM–100 µM	0.1 µM	Rapid, label-free detection, capable of isolating bacteria from whole blood	[100]
Impedance	SPGE immobilized with thiolated vancomycin	HS-Van	*S. aureus*	Vancomycin	10 min	10^1^–10^8^ CFU/mL	<39 CFU/mL	Exhibits high performance for the detection of vancomycin susceptible bacteria	[101]
Impedance, DPV	Gold WE, gold CE, silver RE	Agrose-based hydrogel	*S. aureus*, MRSA	Amoxicillin, oxacillin	<45 min	8 µg/mL and 50 µg/mL	10^4^ CFU/mL	Rapid detection within 45 min, low-cost, commercially available	[102]
DPV	Miniaturized incubation chamber containing WE, CE, and RE	Resazurin	*E. coli, Klebsiella* *pneumoniae*	Ampicillin, ciprofloxacin	30 min	1–1000 CFU/mL	100 CFU/μL	Faster result, effective with clinically relevant levels of bacteria	[21]
DPV	Glassy carbon electrode	Graphene-modified electrode	*E. coli*	Ofloxacin, penicillin, cefepime	30 min	1 × 10^5^ CFU–5 × 10^7^ CFU/mL	10 CFU/mL	Rapid, label-free detection	[103]
DPV	Resazurin modified Graphite SPE	Resazurin	*E. coli*	Gentamycin sulphate	90 min	0–1000 µM	15.6 µM	Rapid, cost-effective	[84]
DPV	SPCE modified with multiwelled carbon tube and gold nanoparticle		*Salmonella gallinarum*	Ofloxacin, penicillin	60 min	10^2^ to 10^7^ CFU/mL	100 CFU/mL	Rapid, highly sensitive	[104]
DPV	Nafion-coated organic redox-active crystal layers on planar pyrolytic graphite sheets		*E. coli*	Ampicillin, kanamycin	60 min	0.001–10 µM, 0 µg/mL, 16 µg/mL	16 µg/mL	Sensors are stable after 60 days of storage in ambient conditions and enable analysis of microbial viability in complex solutions	[105]
DPV, resazurin detection	Platinum WE and CE, Ag/AgCl RE		*E. coli, K. pneumoniae*	Ampicillin, kanamycin, tetracycline	<4 h	10^3^ to 10^8^ CFU/mL	1 × 10^4^ cells/mL	Easy, rapid, reliable, and inexpensive	[106]
Capacitance	Gold electrode on a glass substrate	Aptamer	*E. coli, S. aureus*	Gentamycin	2 h	0–50 µg/mL	10 CFU/mL	Enhanced sensitivity due to AuNP modification	[71]
SWV	mecAgene/MCH/hairpin probe/Au electrode	E-DNA	mecA DNA from MRSA	Methicillin	1–2 h	0–400 pM	63 fM	Hairpin probe design ensures specific binding to the mecA gene	[107]
SWV	E-Si-CRISPR	Aptamer gRNA	mecA DNA from MRSA	Methicillin	60 min	10 fM–0.1 nM	3.5 fM and 10 fM	Amplification-free, ultrasensitive; requires precise gRNA design for different targets	[108]
SWV	Au nanoparticles modified by anti-Pls	MRSA-specific antibody	Antigen	Methicillin	2–3 h	0.2–10 µM 4 × 10^7^–2 × 10^4^ CFU/mL	2 × 10^4^ CFU/mL	Early identification of inflammatory diseases in resistant bacteria	[109]
Amperometry	Screen printed gold electrode (SPGE)	Solid-phase RPA primers	DNA AMR gene of *E. coli*	Oxacillin	60 min	319–20,830 CFU/mL	319 CFU/mL	Rapid detection; direct coupling of the modified RPA reaction to electrochemical measurement simplifies the detection process	[96]
CV	MNP/DNA1- Au/DNA-2	Ferrocene-labeled probes	mecA DNA from MRSA	Methicillin	2 h	10–166 pM	10 pM	Combination of MNPs and AuNPs enhances detection sensitivity Risk of non-specific binding	[110]
CV	TiO2-NTs	PBP2a Protein	*S. aureus*	Methicillin	Few minutes	1–100 ng/mL	1 ng/µL	rapid detection of target antigenic proteins	[111]
CV, DPV	Au/SPCE	Monoclonal anti-MRSA antibody and Aptamer gRNA	Antigen	Methicillin	1–2 h	10–10^6^ CFU/mL	13 CFU/mL	Rapid and label-free detection of highly pathogenic bacteria	[112]
EIS	GCE-APTES-rGOdsDNA	ssDNA	Methicillin-resistant *Staphylococcus aureus* MRSA)	Methicillin	2 h	0.1 pM–1 µM	0.1 pM	Presence of rGO is favored to anchor both ssDNA and dsDNA, which provides the stable response of impedance	[80]
EIS	MCH-sDNA-GE	ssDNA-GE	b-lactam gene	Ampicillin	1 h	3.1–480 pM	3.1 pM	Specific recognition ability for single-base, double-base, and three-base mismatch DNA	[99]
EIS	MSP-TSP/Au electrode	Multisignal Probes	130 nt synthetic ssDNA and gDNA	Methicillin	2.5 h	100 nM–10 fM	10 fM and 57 fM	Specifically, detection of mecA gene, portable on-site detection of MRSA	[56]
EIS	PEI-f-CNT	SATA-8505, bacteriophage	MRSA USA300 strain	Methicillin	30 min	10^2^–10^7^ CFU/mL	1.29 × 10^2^ CFU/mL in blood plasma	Potentially be integrated into a lab-on-a-chip platform for point of care use	[113]
DPV	mecA gene/Au/GCE	mecA gene	mecA DNA from MRSA	Methicillin	<2 h	50–250 pM	23 pM	Successful determination of mecA gene from MRSA	[114]
DPV	UiO-66/BMZIFderived NPCs	ssDNA	mecA and nuc gene DNA from MRSA	Methicillin	2 h	5–1×10^5^ fM	1.6 fM and 3.6 fM	MRSA and SA can be distinguished; great potential in practical applications	[115]
DPV	Au electrode	Monoclonal anti-MRSA antibody	PBP2a antibody	Methicillin	<4.5 h	3–10^5^ CFU/mL	3 CFU/mL	Can easily be modified to capture various bacteria of interest by selecting appropriate capture and detection antibody pairs	[116]
Capacitance	e-AST system on Au	60 aptamers	*E. coli* U433	11 antibiotic drugs	6 h	0.5–128 mg/mL	10^3^ CFU/mL	Rapid AST can increase survival rate of sepsis patients; diagnosis of sepsis by e-AST costly than gold standard method	[117]

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
