# Peer review of "Electrochemical Biosensing for Antibiotic-Resistant Bacteria: Advances, Challenges, and Future Directions"

_micromachines, 2025, doi:10.3390/mi16090986_

Round 1
Reviewer 1 Report
Comments and Suggestions for Authors
Overall, a good article, only minor issues should be addressed before final acceptance.
- Avoid those keywords that have already been used in the title of the MS.
- In the abstract, the word "detect resistance mechanisms" is confusing.
- Where are the positions of Figures 3 and 4 in the MS text?
- Section 3.3 requires more clarity and simplicity for better understanding.
- Add one figure in subsection 3.2.2. for clarity.
Reviewer 2 Report
Comments and Suggestions for Authors
The review titled "Critical Evaluation of Advances, Challenges, and Future Directions in Electrochemical Biosensing Strategies for Antibiotic-Resistant Bacteria" can be accepted for publication after addressing the following comments.
1). Figure 1 is not properly aligned, and some are visible, whereas others are not.
2). In Table 1, headers must be well differentiated and careful formatting of the able is needed; it is not consistent.
3). Some tables lack critical information, such as assay time, linear range, or detection limits for certain detection strategies in Table 1.
4). Usage of abbreviations has to be addressed at first for the better understanding of non-specialist readers.
5). The review does not clearly define its primary objective, making it difficult to discern whether it is intended as a review, a technical guide, or a proposal for future research.
6). While the review discusses innovative technologies, it does not adequately address the practical challenges of implementing these biosensors in clinical or resource-limited settings.
7). There are typographical errors and grammatical inconsistencies throughout the text, which detract from the professionalism of the review.
8). The article focuses on electrochemical methods while neglecting other emerging technologies that could complement or enhance antibiotic resistance detection. This discussion will enhance the readability of the review.
9). Although the conclusion mentions future perspectives, it lacks specific actionable recommendations or a roadmap for advancing the field.
Comments on the Quality of English LanguageLanguage has to be improved.
Reviewer 3 Report
Comments and Suggestions for Authors
This manuscript presents an overview of electrochemical biosensing strategies for detecting antibiotic-resistant bacteria (ARB). The topic is both timely and important, with significant implications for public health diagnostics. However, the current presentation lacks clarity and structural coherence. I recommend acceptance of the manuscript after minor revisions to address the following concerns:
- In the second paragraph of the Introduction, the authors discuss the history of antimicrobial resistance (AMR). To enhance clarity, it is strongly recommended to include a schematic diagram or timeline summarizing the key milestones in AMR development.
- All abbreviations should be defined in full upon first use, with the abbreviation immediately following in parentheses—for example, ARB, AMR, and PCR.
- Many of the representative examples illustrated in the figures are outdated. Please consider replacing them with more recent data to improve the manuscript's novelty and relevance.
- In Figure 3, six types of electrochemical strategies are summarized. Please reorganize their order to align with the sequence and logical flow of the main text.
- Only around 30 of the cited references were published since 2020, accounting for just one-fifth of the total citations. Please update the reference list to include more recent and relevant publications from the past five years.
Round 2
Reviewer 2 Report
Comments and Suggestions for Authors
Accept